# Clinical and Genetic Characterization of Patients with Bartter and Gitelman Syndrome

**DOI:** 10.3390/ijms23105641

**Published:** 2022-05-18

**Authors:** Viviana Palazzo, Valentina Raglianti, Samuela Landini, Luigi Cirillo, Carmela Errichiello, Elisa Buti, Rosangela Artuso, Lucia Tiberi, Debora Vergani, Elia Dirupo, Paola Romagnani, Benedetta Mazzinghi, Francesca Becherucci

**Affiliations:** 1Medical Genetics Unit, Meyer Children’s Hospital, 50139 Florence, Italy; viviana.palazzo@meyer.it (V.P.); samuela.landini@unifi.it (S.L.); rosangela.artuso@meyer.it (R.A.); lucia.tiberi@unifi.it (L.T.); debora.vergani@meyer.it (D.V.); elia.dirupo@meyer.it (E.D.); 2Department of Biomedic, Experimental and Clinical Sciences “Mario Serio”, University of Florence, 50139 Florence, Italy; valentina.raglianti@unifi.it (V.R.); luigi.cirillo@meyer.it (L.C.); paola.romagnani@unifi.it (P.R.); 3Nephrology and Dialysis Unit, Meyer Children’s Hospital, 50139 Florence, Italy; carmela.errichiello@meyer.it (C.E.); elisa.buti@meyer.it (E.B.); benedetta.mazzinghi@meyer.it (B.M.)

**Keywords:** whole-exome sequencing, Bartter syndrome, Gitelman syndrome, salt-losing tubulopathies, genetics, chronic kidney disease

## Abstract

Bartter (BS) and Gitelman (GS) syndrome are autosomal recessive inherited tubulopathies, whose clinical diagnosis can be challenging, due to rarity and phenotypic overlap. Genotype–phenotype correlations have important implications in defining kidney and global outcomes. The aim of our study was to assess the diagnostic rate of whole-exome sequencing (WES) coupled with a bioinformatic analysis of copy number variations in a population of 63 patients with BS and GS from a single institution, and to explore genotype-phenotype correlations. We obtained a diagnostic yield of 86% (54/63 patients), allowing disease reclassification in about 14% of patients. Although some clinical and laboratory features were more commonly reported in patients with BS or GS, a significant overlap does exist, and age at onset, preterm birth, gestational age and nephro-calcinosis are frequently misleading. Finally, chronic kidney disease (CKD) occurs in about 30% of patients with BS or GS, suggesting that the long-term prognosis can be unfavorable. In our cohort the features associated with CKD were lower gestational age at birth and a molecular diagnosis of BS, especially BS type 1. The results of our study demonstrate that WES is useful in dealing with the phenotypic heterogeneity of these disorders, improving differential diagnosis and genotype-phenotype correlation.

## 1. Introduction

Bartter syndrome (BS) and Gitelman syndrome (GS) are rare [1,2] autosomal recessive inherited disorders characterized by hypokalemic metabolic alkalosis, impaired urinary diluting capacity and secondary hyperaldosteronism without hypertension [3]. The primary defect is the genetically-determined functional impairment of specific transporters involved in sodium chloride reabsorption in the thick ascending limb (TAL) of the loop of Henle, or the distal convoluted tubule (DCT), respectively [4], thus, resulting in salt loss, dehydration and acid-base homeostasis perturbation. This phenotypic picture earns the name of salt-losing tubulopathies. The presence of specific clinical features (e.g., childhood or adult onset, hypocalciuria, hypomagnesemia, polyhydramnios leading to premature birth, etc.) represented the base for the clinical classification in subtypes (namely, antenatal and classic BS, Gitelman-like syndromes) [1,5]. However, many studies reported on a wide spectrum of phenotypic severity in all the forms (e.g., BS3 can present antenatal or later in life, with or without hypercalciuria) and accounted for previously undetected rare complications, highlighting how the clinical picture alone can be unreliable in correctly classifying patients [6,7,8,9,10,11]. Given this increasing body of evidence, the clinical classification of salt-losing tubulopathies has recently been coupled to genetic background, linking clinical pictures to specific genetic patterns [5,12,13]. So far, seven genes (namely, *SCL12A1*, *KCNJ1*, *CLCNKA*, *CLCNKB*, *BSND*, *MAGED2*, *SLC12A3*) have been recognized as responsible for BS and GS, accounting for about 70–80% of genetically confirmed cases [1,5,14]. Very recently, mutations in mitochondrial DNA were reported in association with Gitelman-like phenotypes, thus expanding the genetic background of these disorders to additional patterns of inheritance [15]. In recent years, rapidly decreasing costs and turnaround time led next-generation sequencing (NGS) to spread in diagnostics of inherited tubulopathies, including BS and GS [13,16]. Nowadays, genetic testing with NGS is the reference standard for diagnosis and molecular classification of salt-losing tubulopathies, ensuring unparalleled diagnostic yield and allowing clinicians to establish a diagnosis of certainty, to resolve difficult cases, to screen for extrarenal manifestations and to provide genetic counseling [5,12,13,14]. It performs better in children, probably because of the wider spectrum of possible differential non-genetic diagnosis and atypical or insidious presentation in adults [5,12,13,14]. Besides improving diagnosis, the widespread use of genetic testing to confirm the clinical diagnosis unveiled that salt-losing tubulopathies can be phenocopied by other genetic disorders. These include inherited nephropathies (e.g., HNF1B-nephropathy, other tubulopathies, etc.) and genetic diseases outside the spectrum of primary kidney diseases (e.g., congenital chloride diarrhea, cystic fibrosis, etc.) [1,5,12], thus questioning the strength of genotype-phenotype correlations, including the long-term outcome [14,17]. Defining a clear genetic diagnosis represents the base to investigate the prognosis of these disorders, allowing clinicians to collect data on the disease course and to perform accurate follow-up studies spanning even over the pediatric age, when most of the cases are still currently diagnosed [18]. More importantly, reports assessing the prevalence of chronic kidney disease (CKD) in patients with salt-losing tubulopathies are scarce and fragmented, mostly due to the rarity of the diseases and the lack of dedicated long-term assessment. CKD has been reported in patients with BS with extremely variable rates (0–27%), likely because of variable lengths of follow-up and different genotyping strategies [19,20,21]. Conversely, only a few reports accounted for the risk of CKD in patients with GS [22,23]. Nephrocalcinosis, chronic hypokalemia, long-term treatment with non-steroidal anti-inflammatory drugs (NSAIDs) were reported as possible causes of CKD in patients with BS and GS, but results are controversial and not conclusive [21,24]. The aim of this study was to explore genotype-phenotype correlations and CKD development in a population of patients with BS and GS from a single institution by implementing whole-exome sequencing (WES) with a bioinformatic in-house pipeline for analysis of copy number variations (CNVs). We also aimed to show how a correct genetic stratification of patients can result in informing patient prognosis and investigating the pathophysiologic mechanism underlying disease phenotypes and long-term outcome, including CKD.

## 2. Results

### 2.1. Results

#### 2.1.1. Genetic Diagnosis

A total of 68 index patients referred for genetic testing were evaluated for enrollment in the study. After the application of exclusion criteria, 63 patients from 59 families were included. The baseline characteristics of the study population are reported in Table 1. Patients were mostly of self-reported Caucasian ancestry (59/63, 93.7%) and were almost equally distributed between genders (males 30/63, 47.6%). The median age at onset was 6 years (range 0–54 years), while the median age at referral was 11.5 years (range 0–64 years) (Table 1). Of note, 18 patients (28.5%) were older than 16 years at onset. In addition, 19/56 (34%) patients showed a positive familial history of kidney diseases, while consanguinity was reported only in one case (1/56, 1.8%).

During pre-WES consultation, clinical suspicion leading to referral was re-evaluated before ordering genetic testing (see Methods for details). Pre-WES assessment allowed us to reframe the clinical suspect of GS in 13/36 (36%) patients (Figure 1, left side; Appendix A, cases 25–37) referred for BS. Conversely, the clinical suspect of GS was confirmed by clinical evaluation in 19/19 (100%) patients (Appendix A, cases 38–56). Regarding patients referred for BS/GS (8 patients; Appendix A, cases 24 and 57–63), we refined the clinical suspect in BS in one patient and GS in 7 patients (Appendix A, cases 57–63). All the patients underwent WES in order to obtain a molecular diagnosis. We identified causative variants in 54/63 patients (85.7%) (Appendix A). We observed a significant difference in the diagnostic rate between pediatric and adult patients (93.3% vs. 55.6%, *p* < 0.05). The most commonly mutated gene was *SLC12A3* (*n* = 34), followed by *CLCNKB* (*n* = 6) and *SLC12A1* (*n* = 5) (Appendix A). In most patients, bi-allelic variants in autosomal recessive genes were inherited as compound heterozygosity (39 patients), and only 14 patients showed homozygous variants in disease-causing genes (Appendix A). One patient showed a de novo heterozygous variant in the autosomal-dominant *CASR* gene (Appendix A). Most variants were single nucleotide variants (missense *n* = 48, nonsense *n* = 4, splicing *n* = 6, and frameshift *n* = 6). By applying a combination of read depth and in-house bioinformatics pipelines, we were able to identify CNVs in 9 patients (Appendix A), raising diagnostic yield from 71.4 to 85.7%. CNVs were found in *CLCNKB*, *CBLCNKB*/*CLCKNA*, *SLC12A3* and *SLC12A1* (Appendix A). Of note, 24 variants were not previously reported at the time of annotation (Appendix A).

WES allowed us to confirm the clinical diagnosis in 45 of 63 patients (71.4%) (Figure 1, Appendix A). In contrast, genetic testing modified the clinical diagnosis in 9/63 (14.3%) patients. Of note, we obtained disease reclassification in 3/63 (4.8%) patients. In detail, we obtained a conclusive genetic diagnosis in 23/24 (95.8%) and 31/39 (79.5%) patients with the clinical suspicion of BS and GS, respectively (Figure 1, Appendix A). Among patients with a clinical suspicion of BS (*n* = 24), the clinical diagnosis was confirmed in 16/24 (66.7%) patients, allowing improvement in the classification based on the altered gene/s in all 16 patients (i.e., BS 1–4 subtypes) (Figure 1, Appendix A). In this group, WES enabled us to conclude for a molecular diagnosis of GS in 5/24 (20.8%) patients, thereby changing the clinical diagnosis. In two patients (8.3%) we established a diagnosis of cystic fibrosis and Autosomal dominant hypocalcemia with Bartter syndrome (Figure 1, Appendix A), reclassifying the clinical diagnosis. Only one patient was negative. In the group of patients enrolled with a clinical suspicion of GS (*n* = 39), WES confirmed the clinical diagnosis in 29/39 (74.4%) patients (Figure 1, Appendix A). Conversely, the clinical diagnosis was modified in 2/39 (5.1%) patients. In one patient, we established a diagnosis of BS3. In one additional patient, WES allowed us to obtain a diagnosis of HELIX Syndrome. In addition, eight patients were negative (28.5%) (Figure 1, Appendix A).

Taken together, these results suggest that WES, coupled with CNVs detection pipelines, has a high potential to detect a molecular diagnosis in patients presenting with a clinical picture of salt-losing tubulopathies, allowing reframing of the clinical suspect in about 14% of patients.

#### 2.1.2. Clinical and Biochemical Features of Patients According to Genetic Diagnosis

We then analyzed clinical and laboratory features of patients with a molecular diagnosis of BS and GS in order to assess genotype-phenotype correlations. Detailed information is reported in Table 2. Although patients with BS showed a significantly lower age at onset in comparison to patients with GS (2 months vs. 7 years, *p* < 0.001; Table 2, Figure 2A), eight patients with a molecular diagnosis of GS were younger than 3 years at onset and 15 were younger than 6 years (Appendix A). The age at molecular diagnosis was higher than the age at onset in both BS and GS groups, pointing to significant delay in obtaining a conclusive diagnosis (Figure 2A). According to the literature, gestational age and weight at birth were lower in patients with BS in comparison to patients with GS (Table 2, Figure 2B,C). Overall, premature birth and low birthweight (LBW) were observed in 11 and 10 patients, respectively (Appendix A). Of note, patients with GS were all born at term and only one showed LBW (Appendix A). Among patients with BS, term birth and normal birth weight were observed in 4/15 (26.7%) and 6/15 (40%), almost exclusively in patients with BS3. Patients with a genetic diagnosis of BS showed higher serum bicarbonate levels in comparison to patients with GS (32 vs. 28 mmol/L, *p* < 0.05; Table 2). On the other hand, venous blood pH and serum potassium levels were not significantly different (Appendix A–C, Table 2). Despite patients with GS exhibiting significantly lower serum magnesium levels (1.6 vs. 1.95 mg/dL, *p* < 0.001; Appendix A), 25% of patients with BS showed hypomagnesemia (Table 2). In addition, 3 over 4 (75%) of these patients were BS3. Patients with GS showed a higher urinary specific gravity in comparison to patients with BS (1018 vs. 1005, *p* < 0.001; Appendix A, Table 2), suggesting the conservation of the ability to concentrate urine in the former group. We did not observe any significant difference in urinary pH (Appendix A, Table 2). Although patients with BS showed a higher frequency of nephrocalcinosis (8/17, 47.1%), it was observed also in 5/32 (15.6%; *p* < 0.05) of patients with GS (Figure 3A, Table 2). Of note, no patients with BS and nephrocalcinosis had a molecular diagnosis of BS3. Hypercalciuria (defined as urinary calcium-to-creatinine ratio >0.2 mg/dL in adults or total urinary calcium excretion >4 mg/kg/die in children or adults) was observed in 6/11 (54.4%) patients with BS, while it was completely absent in patients with GS (Figure 3B, Table 2). Hypercalciuria was reported only in 4/12 (33.3%) patients with nephrocalcinosis, irrespective of the genetic diagnosis. This frequency was not statistically different from that observed in patients without nephrocalcinosis (Figure 3C). Similarly, the frequency of alkaline venous blood pH (venous blood pH ≥ 7.45), hypomagnesemia (serum magnesium <1.7 mg/dL), alkaline urinary pH (urinary pH ≥ 7.5) and urinary specific gravity did not differ between patients with or without nephrocalcinosis (Figure 3D–G). The only parameter that significantly differed between the two groups of patients was estimated glomerular filtration rate (eGFR) at onset (Figure 3H, *p* < 0.05).

Taken together these results suggest that although some clinical and laboratory features are more commonly observed in patients with BS or GS, a significant overlap does exist. Moreover, our results do not support the role of the most common factors reported in association with nephrocalcinosis.

#### 2.1.3. Chronic Kidney Disease

Finally, we analyzed the long-term kidney outcome of patients with a confirmed molecular diagnosis of BS and GS. We were able to collect information about 35 patients (11 BS, 24 GS), with a median age at last follow-up of 14 years (range 1–64 years). In our cohort, CKD ≥ G2 (defined as eGFR < 90 mL/min/1.73 m^2^) [25,26] was reported in 12 patients with a conclusive molecular diagnosis of BS or GS. On the other hand, 23 patients did not show abnormal kidney function at their last evaluation (Table 3). We observed a significantly higher frequency of CKD in patients with BS than in those affected by GS (72.7% vs. 16.7%; *p* < 0.05) (Appendix A), with a median eGFR at last follow up of 79 mL/min/1.73 m^2^ and 124 mL/min/1.73 m^2^, respectively (Figure 4A; *p* = 0.056). Of note, patients with BS showed a median age at last follow-up of 8.5 years (range 0–30 years), while patients with GS showed a median age of 15.5 years (range 3–64 years; *p* < 0.05). Two patients with BS reached end-stage kidney disease (ESKD) requiring kidney replacement therapy at 11 months and 26 years (cases 10 and 23; Appendix A).

Patients with BS showed a significantly lower eGFR at onset, in comparison to patients with GS (31.5 vs. 118.5 mL/min/1.73 m^2^; *p* < 0.001). By analyzing the time-course of eGFR over time, we observed that patients with BS showed flattened eGFR curves, frequently ranging below the limit of 90 mL/min/1.73 m^2^, even in childhood and adolescence (Figure 4B and Appendix A). Conversely, eGFR was above the normal value of 90 mL/min/1.73 m^2^ in most patients with GS, showing a decline suggesting CKD only after puberty (Figure 4B). This trend was confirmed by analysis in patients with more than three eGFR measures during follow up (Figure 4C,D).

We then analyzed the possible associations between CKD at last follow-up and clinical and laboratory features of patients (Table 3). Interestingly, patients with CKD showed higher age at last follow-up, longer follow-up, lower birth weight and gestational age at birth, lower eGFR at onset in comparison to patients without CKD (Table 3; all *p* < 0.05). We also observed a higher frequency of BS in patients with CKD (66.7% vs. 13%, *p* = 0.004; Table 3, Appendix A). Conversely, other features, including nephrocalcinosis, NSAIDSs therapy and persistent hypokalemia did not show a different distribution (Table 3). At univariate analysis, CKD was associated only with gestational age (OR = 0.64, 95% CI 0.420–0.974; *p* = 0.037) and a genetic diagnosis of BS (OR = 13.33, 95% CI 2.419–73.483; *p* = 0.003) (Table 4). Among patients with CKD, 5/12 (41.7%) had a molecular diagnosis of BS1, while 2/12 (16.7%), 1/12 (8.3%) and 4/12 (33.3%) showed variants in genes responsible for BS3, BS4a and GS, respectively (Appendix A). All patients with BS1 developed CKD, i.e., no patients with normal eGFR at last follow up had BS1 (𝛘^2^ 8.04; *p* = 0.0046). Of note, patients with GS were older at last follow-up, allowing us to rule out a bias dependent on time in outcome evaluation. Interestingly, 3/5 patients with BS1 showed proteinuria. We observed severe functional impact of variants (i.e., complete loss-of-function) in all these three patients.

Taken together, these results suggest that CKD is more frequent in patients with BS than GS, even if it can be observed also in the latter. In our cohort, the features associated with CKD were a lower gestational age at birth and a molecular diagnosis of BS. Indeed, most patients with CKD showed a molecular diagnosis of BS1.

## 3. Discussion

In this study, we reported on the clinical and genetic characterization of a large cohort of patients with salt-losing tubulopathies. We showed that: 1. WES, coupled with CNVs analysis, has a high potential to establish a conclusive diagnosis, either in children or in adults, with a clinical suspicion of salt-losing tubulopathies; 2. WES permits disease reclassification in almost 15% of cases and is pivotal in defining genotype-phenotype correlations; 3. CKD has a prevalence of nearly 30% in patients with salt-losing tubulopathies, with a higher frequency and a younger age at onset in BS than in GS; 4. Nephrocalcinosis, chronic hypokalemia, birth weight and gestational age are unreliable in predicting CKD, which is rather associated with a genetic diagnosis of BS, especially BS1. To the best of our knowledge, this is the first study evaluating all these aspects in a large cohort of patients with genetically confirmed BS and GS.

Nowadays, genetic testing is recognized as the gold standard for the diagnosis of tubulopathies, including BS and GS [1,5]. In the present study, we provided evidence that WES, coupled with CNVs bioinformatic detection pipelines, has a high potential to detect a molecular diagnosis in patients presenting a clinical picture of salt-losing tubulopathies, reaching a diagnostic rate as high as 86%, substantially higher than previously reported [14,27,28]. Our strategy allowed us to identify CNVs in 17% of patients with a conclusive molecular diagnosis thus increasing the diagnostic yield by an additional 14% (from 71% to 86%). Large rearrangements were detected in genes other than *CLCNKB* and *CBLCNKB*/*CLCKNA*, suggesting that a non-targeted approach for CNVs detection is helpful in unraveling the molecular diagnosis of these disorders. According to previous studies [14,27,28], we obtained a high diagnostic rate (above 90%) in patients with childhood-onset salt-losing tubulopathies, supporting the hypothesis that these disorders show a solid genetic base and that most causative genes have already been identified. However, in our cohort nearly one third of patients were older than 16 years at clinical presentation. In this age group, we achieved a molecular diagnosis in more than 50% of cases, sustaining the need of investigating the genetic background lining behind metabolic abnormalities suggesting salt-losing tubulopathies also in the adult population, especially when secondary causes are ruled out. Of note, inherited tubulopathies, including BS and GS, have recently been demonstrated to represent the cause of a subset of adult patients with CKD of unknown origin [9,29,30,31], further supporting the need to extend genetic testing for tubulopathies also in this group. In this view, the progressive decline in costs and turnaround time will probably support WES as a useful tool in the diagnostic strategy of salt-losing tubulopathies, redefining currently set age boundaries.

Genomic medicine is increasingly promoting personalized approaches to define diagnostic trajectories of inherited kidney diseases [30,32]. In patients with salt-losing tubulopathies, precision medicine claims to build up strategies to cover diagnostic and phenotypic areas of uncertainty in order to inform treatment, prognosis and counseling. Reverse genetics (i.e., reframing clinical diagnosis in view of genetic results) and identification of phenocopies have already proved efficacious in guiding this process in other fields of nephrology (e.g., podocytopathies and steroid-resistant nephrotic syndrome, ciliopathies, congenital anomalies of the kidney and urinary tract, etc.) [33,34,35], paving the way for a profound revision of disease ontology. Previous studies reported on disease reclassification based on reverse genetics, including in tubulopathies [14,18,28,36]. In our work, we were able to carefully assess this aspect by evaluating the clinical and laboratory picture of patients before and after genetic testing, optimizing genetic results interpretation (i.e., providing the laboratory with accurate phenotypic information to guide variant prioritization, particularly regarding the pattern of inheritance and the clinical significance). This multi-step process allowed us to modify about 14% of clinical suspects, showing that clinical diagnosis can be deceptive, even when formulated by experts [10,13]. Indeed, the clinical overlap prevents clinicians from overseeing the genotype from the clinical presentation. In our cohort, age at onset, birthweight, gestational age, hypomagnesemia and nephrocalcinosis failed to predict genetic diagnosis in 15-40% of patients. In particular, nearly half of patients with a molecular diagnosis of GS were younger than six years old at clinical onset, with eight patients presenting before three years, virtually excluding this genetic diagnosis on a clinical basis [1]. These observations suggest that age is not an absolute indicator of the underlying molecular diagnosis, especially of GS. Similarly, nephrocalcinosis is considered a strong indicator of BS. By contrast, in our cohort, we found nephrocalcinosis in three patients with genetically confirmed GS. Since additional genetic factors potentially influencing this phenotypic trait have been ruled out by WES, the results of this study suggest that nephrocalcinosis should not be considered a discriminating factor to clinically distinguish BS from GS, and that the pathogenic mechanism responsible for nephrocalcinosis needs to be further elucidated.

Finally, the most important observations of this study are represented by the long-term characterization of BS and GS outcomes, particularly the prevalence, and possible cause, of CKD. CKD represents a potential complication of salt-losing tubulopathies, especially BS [20,21,37]. However, prevalence and etiology are still poorly defined [5,6,38]. In our cohort, CKD was reported in 34% of patients with a molecular diagnosis. Interestingly, patients with BS showed a high prevalence of CKD (72.7%), usually developing earlier in the disease course in comparison to patients with GS, who showed CKD in nearly 17% of cases and only after puberty. This is in line with the observation that eGFR at onset is lower in patients with BS than in patients with GS, suggesting that features intrinsic to the disease may be involved in the pathogenesis of kidney function decline. Nephrocalcinosis, treatment with NSAIDs, chronic hypokalemia, and prematurity are reported as risk factors for CKD in salt-losing tubulopathies [1,5,6,38]. Secondary renin-angiotensin system (RAS) activation was also proposed as a causal mechanism for kidney impairment [21,24]. In our cohort, most of these features did not show significant association with CKD, which was conversely strongly associated with prematurity and with genetic diagnosis of BS, in particular BS1. Consistently, patients with GS showed a lower risk of CKD, even if they were older than patients with BS at last follow-up, thus ruling out the potential bias of underestimating the prevalence of CKD because of shorter observation time. Although the exact mechanisms of kidney function decline still must be elucidated, the results of our study suggest that pathogenic clues are related to the disease itself. Interestingly, in addition to TAL, the Na^+^:K^+^:2Cl^−^ co-transporter (NKCC2) encoded by *SLC12A1* is expressed in the apical membrane of the macula densa [39,40], the structure responsible for tubulo-glomerular feedback [41]. Functional impairment of the macula densa finally results in tubulo-glomerular feedback inactivation, altering response of the afferent arteriole to changes in intraglomerular pressure [41]. These results suggest that in patients with BS1 tubulo-glomerular feedback is non-functional, due to genetic abnormalities in *SLC12A1*, leading to a strong activation of RAS, glomerular hyperfiltration and, finally, to kidney function impairment [5].

This study has some limitations. Firstly, our population is relatively small and geographically restricted, with a limited inclusion of patients of non-Caucasian ancestry. However, our cohort is comparable with those previously reported in multi-centric studies and it is, to our knowledge, the largest from a single institution [6,14,18,21,42]. Secondly, the retrospective nature of the study, together with the monocentric design, limited data completeness. This surely affects the robustness of data that needs to be further assessed in larger multi-center cohorts and prospective studies. Finally, the mechanisms hypothesized for CKD development need experimental verification. Consequently, additional studies are required to address unsolved issues.

In conclusion, in this study we reported on phenotypic and genetic characterization of a large cohort of patients with BS and GS with long-term follow-up data at a single tertiary center. We showed that WES, coupled with bioinformatic analysis for CNVs, leads to provision of high diagnostic yields, either in children or in adults, allowing clinicians to confirm the clinical suspect or to revise it on the basis of genetic results. Disease reclassification occurs at a not negligible frequency in patients with salt-losing tubulopathies, confirming that the clinical picture can mislead diagnostic trajectories. In our cohort, about one third of patients with BS and GS showed CKD, suggesting that the prognosis of these disorders is all but benign in the long run. The genetic diagnosis, and particularly BS1, seems to represent the strongest risk factors for CKD, suggesting that assessing long-term prognosis in genetically stratified cohorts of patients is mandatory to correctly define patient prognosis and optimize management.

## 4. Materials and Methods

### 4.1. Patients

Patients with suspected salt-losing tubulopathies (i.e., BS, GS or BS/GS spectrum) were referred for clinical evaluation and molecular diagnosis from peripheral nephrology units to the tertiary center Meyer Children’s Hospital of Florence (Italy). Referral was ordered by the treating nephrologist or pediatrician because of persistent metabolic alkalosis and/or hypokalemia, after the exclusion of secondary causes of the clinical phenotype (e.g., diuretics, laxative consumption, surreptitious drugs assumption, anorexia, autoimmune disease, etc.) [1,5]. At the tertiary center, patients were evaluated by a nephrologist with expertise in genetic kidney diseases and certified for genetic counseling, as well as by a clinical geneticist at the “Rare Genetic Kidney diseases” outpatient service. Patients were evaluated at the tertiary center whenever possible. Anyway, in case of strong clinical suspicion from the referral clinicians, patients were included in the study. During pre-WES consultation, all relevant clinical information was collected. Demographic, clinical, instrumental and laboratory data were retrospectively collected from direct interview of patients, families, and referring clinicians and from medical records. Comprehensive phenotypic information, and results of other genetic testing, if previously performed, were collected. All the information collected during pre-WES consult was used to reframe the clinical suspect of BS or GS. All consecutive patients referred from 2016 to December 2021 were evaluated for enrollment in the study. Genetic testing with WES was offered to all the patients enrolled. Parents were either included before, or asked to participate after, the identification of potentially causative variants in the proband, in order to assess variant segregation. All the patients, or their legal representatives, were counseled regarding the WES procedure and gave written informed consent for genetic testing before blood sampling. The local Ethics Committee of the Meyer Children’s University Hospital of Florence approved the study (Approval n. 50/2020). The study was conducted according to the Declaration of Helsinki.

### 4.2. Whole-Exome Sequencing and Bioinformatic Analysis

Genomic DNA was extracted from peripheral blood using QIAamp Mini Kit (QIAGEN^®^, Hilden, Germany). Libraries were constructed with enzymatic fragmentation followed by End repair, A-tailing, adapter ligation and library amplification (KAPA HyperPlus Kits, Roche, Basel, Switzerland). Libraries were hybridized to the whole-exome capture arrays (SeqCap EZ Exome v3, Nimblegen, Roche, Basel, Switzerland) and sequenced with NextSeq500/550 (Illumina Inc., San Diego, CA, USA). The reads were aligned with the human reference hg19 genome using Burrows-Wheeler Aligner (BWA)), mapped and analyzed with the IGV software (Integrative Genome Viewer, 2013 Broad Institute and Regents of the University of California, CA, USA) [43]. Downstream alignment processing was performed with the Genome Analysis Toolkit Unified Genotyper Module (GATK) [44], SAMtool [45] and Picard Tools Variants were annotated using Annovar tool [46] to obtain information such as variant frequency in different populations and the predictions of the variant effect using different softwares (SIFT, Polyphen2, MutationTaster, MutationAssessor, FATHMM and FATHMM MKL). We retained non-synonymous, short insertion/deletion, synonymous or splice-site variants (20 bp splice acceptor, 20 bp splice donor) with the following characteristics: -Variants in genes described in Online Mendelian Inheritance in Man (OMIM) and/or Human Gene Mutation Database (HGMD) or in scientific literature revised at 28 February 2022; Variants not present or with a minor allele frequency (MAF) ≤ 0.01 for autosomal recessive and with a MAF ≤ 0.001 for autosomal dominant -transmitted genes in population database “1000Genomes Project”, “Exome Variant Server”, dbSNP153, and in our “in-house” exome control cohort (2000 exomes) of unrelated subjects analyzed for non-renal diseases referred to the Medical Genetics Unit of the Meyer University Hospital (Florence, Italy); -Variants not present in the database of healthy control populations (gnomAD) in homozygous or in hemizygous state; -Variants reported in disease-causing mutations databases as ClinVar, HGMD or Decipher v11.7, or predicted as damaging by at least 3 in silico tools (Polyphen-2, SIFT, Mutation Taster, MutationAssessor, FATHMM, FATHMM MKL). For splicing and synonymous variants, we retained only those predicted as causative of splicing alteration by Berkeley Drosophila Genome Project (BDGP). In order to estimate copy number variations (CNVs) we used a normalized read count approach implemented in-house [33]. Selected variants were classified in agreement with the interpretation guidelines of the American College of Medical genetics and Genomics (ACMG) [47] as resulting from Varsome and we retained only those classified as “pathogenic”, “likely pathogenic” or “variant of unknown significance”. According to the functional effect, we classified identified variants as partial loss-of-function (missense variants) or complete-loss-of function (splicing, nonsense, frameshift variants and large deletions). After the selection, the candidate variants were validated in patients and relative samples by Sanger sequencing (for point substitutions) or real time PCR or Array-CGH (for CNVs). We considered only variants in accordance with the pattern of inheritance. When the segregation of the variants was not possible due to the absence of parents’ DNA samples, the variants were retained only in the presence of clear genotype–phenotype correlation.

### 4.3. Statistical Analysis

Statistical analysis was performed using SPSS software v28.0 (SPSS, Inc., Evanston, IL, USA). Continuous variables are shown as median and range. Between groups differences were evaluated through Student’s *t* test or Mann-Whitney U test, as appropriate. Categorical variables are expressed as absolute frequencies and percentages and were compared by the Chi-square test with Yates correction. Logistic unadjusted regression models were used to estimate the risk of CKD (eGFR at last available follow up < 90 mL/min/1.73 m^2^, according to baseline demographic and clinical parameters, and results were reported as crude odds ratios (ORs) with 95% confidence intervals (95% CIs). The level of statistical significance was set at a value of *p* < 0.05.

### 4.4. Web Resources

1000 Genomes Project, http://www.1000genomes.org, accessed on 28 February 2022.

Exome Variant Server of the NHLBI Exome Sequencing Project (ESP), http://evs.gs.washington.edu/ews, accessed on 28 February 2022.

dbSNP153, https://www.ncbi.nlm.nih.gov/projects/SNP/, accessed on 28 February 2022.

gnomAD, http://gnomad.broadinstitute.org/, accessed on 28 February 2022.

Human Gene Mutation Database (HGMD), http://www.hgmd.cf.ac.uk/ac/index.php, accessed on 28 February 2022.

Clinvar database, http://www.ncbi.nlm.nih.gov/clinvar/, accessed on 28 February 2022.

Polyphen-2, http://genetics.bwh.harvard.edu/pph2/, accessed on 28 February 2022.

SIFT, http://sift.jcvi.org/, accessed on 28 February 2022.

Mutation Taster, http://www.mutationtaster.org/, accessed on 28 February 2022.

Mutation Assessor, http://www.ngrl.org.uk/Manchester/page/mutation-assessor, accessed on 28 February 2022.

FATHMM, http://fathmm.biocompute.org.uk/, accessed on 28 February 2022.

FATHMM MKL, http://fathmm.biocompute.org.uk/fathmmMKL.ht, accessed on 28 February 2022.

Berkeley Drosophila Genome Project (BDGP), https://www.fruitfly.org, accessed on 28 February 2022.

Online Mendelian Inheritance in Man (OMIM), https://www.omim.org, accessed on 28 February 2022.

Decipher v11.7, https://www.deciphergenomics.org/, accessed on 28 February 2022.

Varsome, https://varsome.com/, accessed on 28 February 2022.

Picard Tools, http://picard.sourceforge.net/, accessed on 28 February 2022.

Burrows-Wheeler Aligner, https://sourceforge.net/projects/bio-bwa/files/, accessed on 28 February 2022.

## Figures and Tables

**Figure 1 ijms-23-05641-f001:**
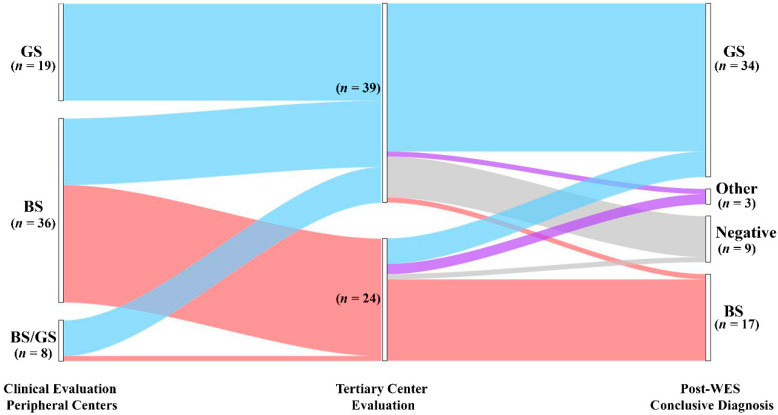
Diagnostic yield and disease reclassification. Patients are initially classified based on the clinical suspicion raised by referring physicians (left node). The clinical diagnosis was reframed during pre-WES consultation (middle node). All the patients underwent genetic testing with WES that allowed us to confirm or reclassify the clinical diagnosis (right node). GS, Gitelman syndrome; BS, Bartter syndrome; Other: genetic diagnosis outside BS and GS; Negative: patients without pathogenic variants.

**Figure 2 ijms-23-05641-f002:**
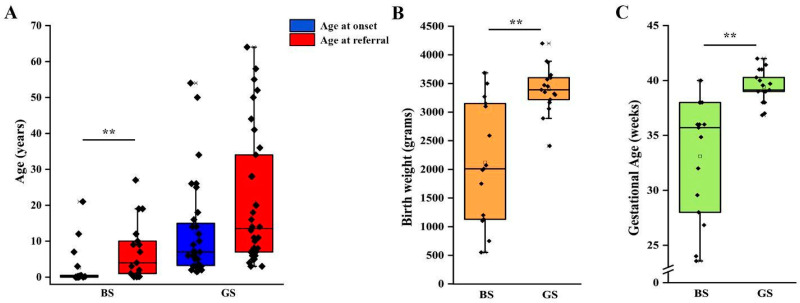
Clinical features of patients according to genetic diagnosis. (**A**) Age at onset and age at referral for genetic testing in patients with a molecular diagnosis of BS and GS. (**B**) Birth weight in patients with a molecular diagnosis of BS and GS. (**C**) Gestational age at birth in patients with a molecular diagnosis of BS and GS. Individual scores are shown as ◆. ** *p* value < 0.001. GS, Gitelman syndrome; BS, Bartter syndrome.

**Figure 3 ijms-23-05641-f003:**
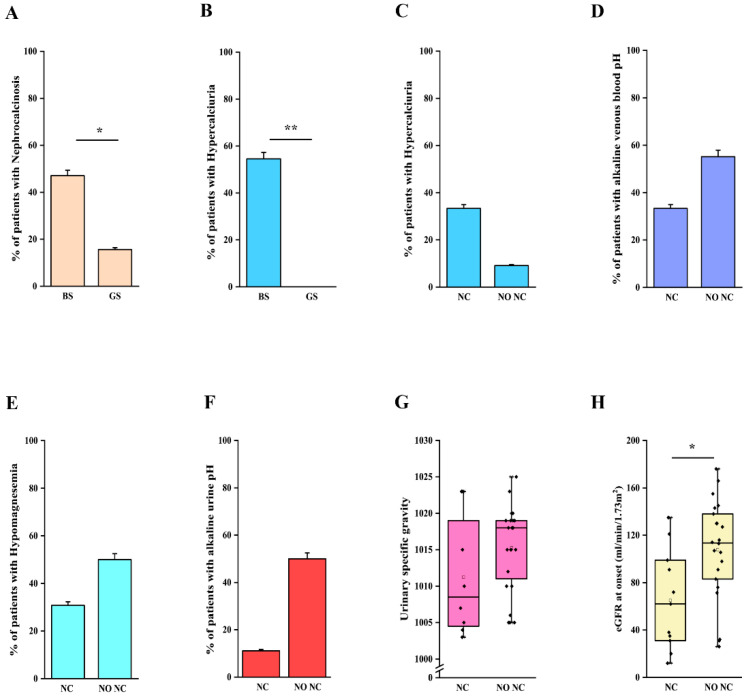
Clinical and laboratory features in patients with nephrocalcinosis. (**A**) Frequency of nephrocalcinosis in patients with a molecular diagnosis of BS and GS. (**B**) Frequency of hypercalciuria in patients with a molecular diagnosis of BS and GS. (**C**) Frequency of hypercalciuria in patients with and without nephrocalcinosis, irrespective of the molecular diagnosis. (**D**) Frequency of alkaline venous blood pH (venous blood pH ≥ 7.45) in patients with and without nephrocalcinosis, irrespective of the molecular diagnosis. (**E**) Frequency of hypomagnesemia (serum magnesium < 1.7 mg/dL) in patients with and without nephrocalcinosis, irrespective of the molecular diagnosis. (**F**) Frequency of alkaline urinary pH (urinary pH ≥ 7.5) in patients with nephrocalcinosis, irrespective of the molecular diagnosis. (**G**) Distribution of urinary specific gravity in patients with and without nephrocalcinosis, irrespective of the molecular diagnosis. (**H**) Distribution of eGFR in patients with and without nephrocalcinosis, irrespective of the molecular diagnosis. eGFR was calculated with Schwartz revised formula in children and CKD-EPI in adults. Individual scores are shown as ◆. * *p* value < 0.05, ** *p* value < 0.001. GS, Gitelman syndrome; BS, Bartter syndrome; NC, nephrocalcinosis; NO NC, absence of nephrocalcinosis; eGFR, estimated glomerular filtration rate.

**Figure 4 ijms-23-05641-f004:**
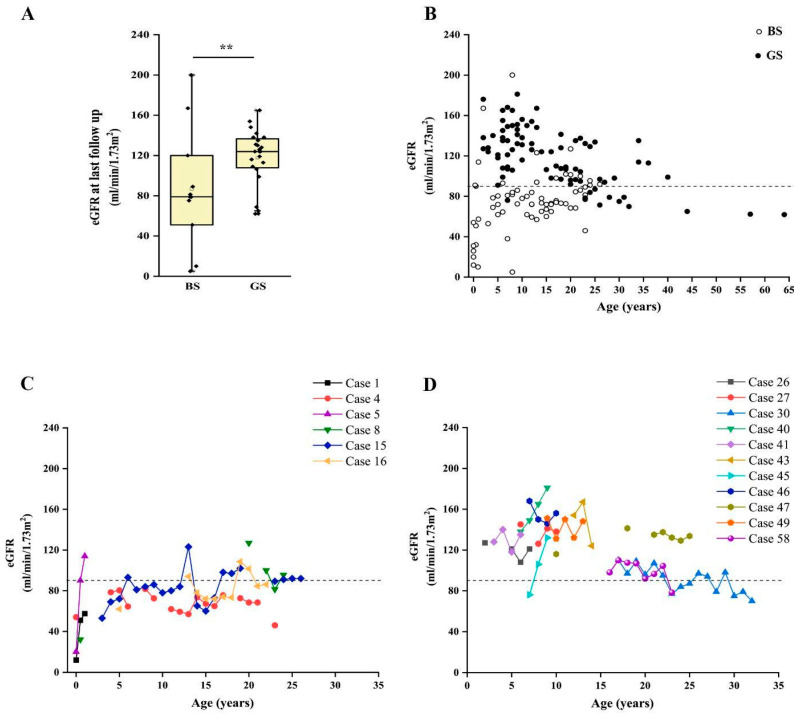
Long-term outcome of patients with BS and GS. (**A**) Box and whiskers plot of eGFR at last follow up in patients with a molecular diagnosis of BS and GS. Individual scores are shown as ◆. (**B**). Plot of eGFR over time in patients with a molecular diagnosis of BS (white dots) and GS (black dots). Each dot represents a measurement in one patient. (**C**) Plot of the eGFR over time in patients with a molecular diagnosis of BS with at least three measurements during follow up. (**D**) Plot of the estimated glomerular filtration rate over time in a subset of patients with a molecular diagnosis of GS with at least three measurements during follow up. eGFR was calculated with Schwartz revised formula in children and CKD-EPI in adults. *** p* value < 0.001. GS, Gitelman syndrome; BS, Bartter syndrome; eGFR, estimated glomerular filtration rate.

**Table 1 ijms-23-05641-t001:** Summary of clinical and biochemical features of the patients included in the study.

	Population (*n* = 63)
Male, n (%)	30 (47.6)
Caucasian, n (%)	59 (93.7)
Age at onset, years (median, range)	6 (0–54)
Age at last follow up, years (median, range)	17 (0–64)
Length of follow up, years (median, range)	3 (0–37)
Consanguinity, n (%)	1/56 (1.8)
Family history, n (%)	19/56 (33.9)
Polyhydramnios, n (%)	14/57 (24.6)
Gestational age at birth, weeks (median, range)	38 (23–42)
Birthweight, grams (median, range)	3194 (550–4200)
Nephrocalcinosis, n (%)	16/61 (26.2)
Failure to thrive, n (%)	23/62 (37.1)
Polyuria, n (%)	10/39 (25.6)
SNHL, n (%)	1/58 (1.7)
Venous blood pH, (median, range)	7.45 (7.35–7.61)
Serum bicarbonate, mmol/L (median, range)	28.35 (24–43)
Serum potassium, mmol/L (median, range)	2.6 (1.40–3.80)
Serum sodium, mmol/L (median, range)	137 (124–143)
Serum chloride, mmol/L (median, range)	96 (56–105)
Total serum calcium, mg/dL (median, range)	9.6 (4–11.6)
Serum magnesium, mg/dL (median, range)	1.7 (0.9–2.9)
Hypomagnesemia, n (%)	25/56 (44.6)
Hyperaldosteronism, n(%)	31/50 (62)
Hyperreninemia, n (%)	32/45 (71.1)
Serum Creatinine, mg/dL (median, range)	0.62 (0.20–1.56)
eGFR at onset, mL/min/1.73 m^2^ (median, range)	105.5 (12–176)
eGFR < 90 mL/min/1.73 m^2^ at onset, n (%)	15/41 (36.6)
Urinary pH at onset (median, range)	7 (5.5–8)
Urinary specific gravity at onset (median, range)	1015 (1003–1025)
Hypercalciuria, n (%)	9/43 (20.9)

SNHL, sensorineural hearing loss; eGFR, estimated glomerular filtration rate.

**Table 2 ijms-23-05641-t002:** Clinical and biochemical features of patients according to genetic diagnosis.

	BS (*n* = 17)	GS (*n* = 34)	*p*-Value
Male, n (%)	5 (29.4)	23 (67.6)	0.022
Caucasian, n (%)	16 (94.1)	31(91.2)	-
Age at onset, years (median, range)	0.16 (0–21)	7 (2–54)	<0.001
Age at last follow up, years (median, range)	8.5 (0–30)	15.5 (3–64)	0.023
Length of follow up, years (median, range)	1.67 (0–30)	3.5 (0–34)	0.606
Consanguinity, n (%)	1/16 (6.3)	0/29 (0)	0.76
Family history, n (%)	5/16 (31.3)	12/30 (40)	0.79
Polyhydramnios, n (%)	12 (70.6)	1/29 (3.4)	<0.001
Gestational age at birth, weeks (median, range)	35 (23–40)	39 (37–42)	<0.001
Birth weight, grams (median, range)	2010 (550–3685)	3390 (2410–4200)	<0.001
Nephrocalcinosis, n (%)	8 (47.1)	5/32 (15.6)	0.042
Failure to thrive, n (%)	10/16 (62.5)	12 (35.3)	0.133
Polyuria, n (%)	7/10 (70)	1/25 (4)	<0.001
SNHL, n (%)	1 (5.9)	0/29 (0)	0.785
Venous blood pH, (median, range)	7.5 (7.37–7.61)	7.44 (7.35–7.56)	0.096
Serum bicarbonate, mmol/L (median, range)	32 (25.4–39)	28 (24–34)	0.015
Serum potassium, mmol/L (median, range)	2.6 (1.4–3.6)	2.7 (1.5–3.5)	0.887
Serum sodium, mmol/L (median, range)	133 (124–142)	137 (133–143)	0.043
Serum chloride, mmol/L (median, range)	92 (56–102)	98 (85–105)	0.012
Total serum calcium, mg/dL (median, range)	9.7 (4–11.6)	9.6 (8.5–11)	0.927
Serum magnesium, mg/dL (median, range)	1.95 (1.2–2.9)	1.6 (1.1–2)	<0.001
Hypomagnesemia, n (%)	4/16 (25)	16/29 (55.2)	0.102
Hyperaldosteronism, n(%)	12/14 (85.7)	13/24 (54.2)	0.105
Hyperreninemia, n (%)	13/13 (100)	14/21 (66.7)	0.057
Serum Creatinine, mg/dL (median, range)	0.7 (0.37–1.56)	0.45 (0.2–1.35)	0.006
eGFR at onset, mL/min/1.73 m^2^ (median, range)	31.5 (12–91)	118.5 (34–176)	<0.001
eGFR < 90 mL/min/1.73 m^2^ at onset, n (%)	9/10 (90)	3/22 (13.6)	<0.001
Urinary pH at onset (median, range)	7 (5.5–8)	7 (5.5–8)	0.749
Urinary specific gravity at onset (median, range)	1005 (1003–1010)	1.18 (1006–1025)	<0.001
Hypercalciuria, n (%)	6/11(54.5)	0/23 (0)	<0.001

BS, Bartter syndrome; GS, Gitelman syndrome; SNHL, sensorineural hearing loss; eGFR, estimated glomerular filtration rate.

**Table 3 ijms-23-05641-t003:** Descriptive analysis of clinical and laboratory features of patients with or without CKD and molecular diagnosis of BS or GS.

	CKD (*n* = 12)	Non CKD (*n* = 23)	*p*-Value
Genetic diagnosis (BS vs. GS), n (%)	8 (66.7)	3 (13)	0.004
Functional impact (CL vs. PL), n (%)	8 (66.7)	12 (52.1)	0.64
Gender (male vs. female), n (%)	6 (50)	13 (56.5)	0.99
Ethnicity (caucasian vs. other), n (%)	12 (100)	21 (91.3)	0.77
Age at diagnosis, years (median, range)	1.75 (0–54)	6 (0–34)	0.234
Age at last follow up, years (median, range)	25 (1–64)	10 (1–40)	0.041
Length of follow up, years (median, range)	21 (1–34)	3 (0–27)	0.016
Birth weight, grams (median, range)	2070 (550–3500)	3310 (1990–4200)	0.035
Gestational age at birth, weeks (median, range)	35 (24–40)	39 (35–42)	0.011
eGFR at onset, mL/min/1.73 m^2^ (median, range)	34 (12–72)	115 (20–176)	0.001
Hypertension, n (%)	2/11 (18,2)	1/20 (5)	0.58
Proteinuria, n (%)	4/8 (50)	2/20 (10)	0.069
Nephrocalcinosis, n (%)	6/10 (60)	6 (26.1)	0.14
Other renal/urological findings, n (%)	3/1 (25)	2/22 (9)	0.45
NSAIDs, n (%)	5/11 (45.5)	3/20 (15)	0.070
Length of NSAIDs therapy, years (median, range)	23 (11–26)	15 (2–16)	0.143
Persistent hypokalemia, n (%)	5/11 (45.5)	6 (26.1)	0.46

CKD, chronic kidney disease; BS, Bartter syndrome; GS, Gitelman syndrome; CL, complete loss-of-function; PL, partial loss-of-function; eGFR, estimated glomerular filtration rate; NSAIDs, non-steroidal anti-inflammatory drugs. We defined hypokalemia as serum potassium levels < 3 mmol/L; Hypertension as systolic and/or diastolic blood pressure > 140/90 mmHg in adults and > 90th percentile for age and sex in children and adolescence; CKD as eGFR < 90 mL/min/1.73 m^2^.

**Table 4 ijms-23-05641-t004:** Univariate analysis for CKD (crude odds ratio).

	OR (95% CI)	*p*-Value
Genetic diagnosis (ref. BS)	13.33 (2.42–73.48)	0.003
Genetic diagnosis (ref. GS)	0.075 (0.014–0.413)	0.003
Functional impact (ref. CL)	1.833 (0.429–7.836)	0.413
Birth weight (grams)	0.999 (0.998–1.000)	0.018
Gestational age at birth, (weeks)	0.64 (0.42–0.974)	0.037
Nephrocalcinosis	4.25 (0.884–20.44)	0.071
Serum potassium levels (mmol/L)	1.253 (0.378–4.15)	0.713

CKD, chronic kidney disease; BS, Bartter syndrome; GS, Gitelman syndrome; CL, complete loss-of-function; CI, confidence interval.

## Data Availability

Not applicable.

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
