# Peer review of "Clinical and Genetic Characterization of Patients with Bartter and Gitelman Syndrome"

_ijms, 2022, doi:10.3390/ijms23105641_

Round 1

Reviewer 1 Report

The manuscript must be revised, because it does not bring anything innovative. The rationale is not well explained. The introduction should be expanded. Furthermore, the bibliography must be revised, as there is no correspondence between the notes in the article and the references reported. In addition, other references have been added in the supplementary material. The discussion should be expanded to included a weaknesses paragraph. The number of the ethics committee is missing. Finally, the paper is very difficult to read with multiple English grammar errors.

Author Response

We would like to thank the reviewer for sharing these comments that gave us the opportunity to improve our work. We expanded the introduction to better highlight the rationale of the study and emphasise our findings (see page 2 lines 69-75, page 2 lines 91-98, page 3 lines 99-111). We also enriched the paragraph concerning weakness and limitations of our study in the discussion section (see page 14 lines438:442-445). Furthermore, we have better highlighted the innovative observations provided by our study. In particular: 1.  WES coupled with CNVs analysis leads to high diagnostic yields either in children or in adults with a clinical suspicion of salt-losing tubulopathies (see page 12 lines 332-334); 2. The clinical picture of patients can mislead diagnostic trajectories, making genetic testing essential in defining a conclusive diagnosis (see page 12 lines334-336); 3. CKD is a frequent complication of salt-losing tubulopathies (about one third of patients); the genetic background represents the strongest risk factor for CKD in this population (see page 12 lines338-342). To the best of our knowledge, this is the first study evaluating all these aspects in a large cohort of patients with genetically confirmed BS and GS. 

Bibliography also was revised, to check incomplete/incorrect information. The references reported in the Supplementary file concern Table S2; we prefered to keep them separated to the main text to favour readability.

We carefully revised grammar errors and typos, correcting all of them. In addition, the manuscript was revised by an English native speaker.

Reviewer 2 Report

In manuscript titled "Clinical and genetic characterization of patients with Bartter and Gitelman syndrome", Palazzo et al employ a combination of hWES and bioinformatics to enhance genotype-phenotype correlation and aid differential diagnosis in a phenotypically heterogenous disease group.

While the findings are interesting, I find them to be of minimal significance. It is obvious that data derived from WES will surpass those derived from other methods. 

Author Response

We thank the reviewer for the comments. According to the points raised, we reviewed the manuscript to better highlight our intent. In this work we did not aim at demonstrating the superiority of WES over other methods of genetic investigation, since we agree with the reviewer that WES has indisputable stronger power to highlight deeper genotyping and will probably represent the first-choice strategy for genetic investigation in inherited disorders in the next future. Indeed, in our centre we adopted WES and bioinformatic analysis coupled with deep phenotyping before and after genetic variant identification, either to improve diagnosis or to enhance accurate and reliable genotype-phenotype correlations. This is particularly relevant for rare diseases such as BS and GS, for whom data are frequently sparse due to low prevalence. The aim of our work is to underlie how the correct genetic stratification of patients is pivotal in informing genotype-phenotype correlation and investigating disease phenotypes and long-term outcome, including CKD. Despite the limitations stated, we believe that our study represents a good description of a large cohort of patients affected by these rare tubulopathies, providing important information about the prognosis of the disease and warning about the risk of CKD of these patients who are often lost at follow up, especially at the transition to adult nephrology centres. We modified the manuscript according to these suggestions in order to better clarify the aim of the study and the importance of our findings (see page 13 lines 381-383; page 14 lines 405-407).

Reviewer 3 Report

Comments:

  1. Please include one paragraph in the introduction to clarify the aim of your study.
  2. Figure 1 is hard to understand please draw in a different way.
  3. Please add the error bar to Figure 3 (A-F).
  4.  Rewrite the discussion to reflect your findings.

Author Response

We kindly thank the reviewer for the comments and suggestions. We revised the manuscript either in the introduction or in the discussion, in order to better clarify the aim of our study and emphasise our findings (see page 2 lines 91-98; page 3 lines 99-111 and page 12 lines 332-342). The figures have been revised as suggested.

Reviewer 4 Report

Bartter (BS) and Gitelman syndrome (GS) are inherited tubulopathies with challenges in their precise diagnosis since both pathologies have overlapping clinical symptoms. In this paper the authors demonstrated that WES is useful in dealing with the  phenotypic heterogeneity of these disorders, improving differential diagnosis and genotype-phenotype correlation. In particular  the aim of this study was to assess the diagnostic rate of WES coupled with a bioinformatic in house pipeline for analysis of  CNVs in a population of patients with BS and GS from a single institution.

Although the data were well presented, this reviewer request some explanation (which I consider minor points)

Minor points

  • During pre-WES consultation, clinical suspicion leading to referral was re-evaluated before ordering genetic testing (see Methods for details). Pre-WES assessment allowed the authors to  reframe the clinical suspect of GS in 13/36 (36%) patients referred for BS. Conversely, the clinical suspect of GS was confirmed by clinical evaluation in 19/19 (100%) patients . In patients referred for BS/GS, they refined the clinical suspect in BS in one patient and GS in 7 patients. Because  the Authors performed WES analysis, they have the possibility to screen for all causative genes, so the question is: Why they re-evaluated the clinical suspicion before performing genetic testing?
  • In other words, they coud have approached by “reverse phenotyping” (it seems that this is the approach then)

  • The process of searching for pathogenic mutations generally begins with the sequencing of the known associated genes by NGS technologies. The GS and BS genes are often part of predesigned panels of known genes involved in tubulopathies, chronic hypokalemia or salt-acid disorders.

What was the approach of the authors? And did they evaluate the mitochondrial genome?

  • Another complementary method is the search for CNVs (copy number variations) in candidate genes, through specific NGS products and software, arrays of SNPs (single nucleotide polymorphism) or MLPA®(Multiplex Ligation-dependent Probe Amplification), among others. The authors investigated CNV through  NGS AND in house bioinformatic pipeline.

Did they confirm these CNV? B SNP Array or MLPA?

  • Can the authors add some comments about the unsolved cases? . Investigation of unresolved cases and their parents using whole genome strategies could lead to the discovery of new genes associated with GS or BS; or (d) the identification of a second mutation is not described since it does not really exist.

Author Response

We thank the reviewer for the suggestions that give us the opportunity to clarify some issues and to improve our manuscript. Please find below a point-by-point discussion of the concerns raised: 

During pre-WES consultation, clinical suspicion leading to referral was re-evaluated before ordering genetic testing (see Methods for details). Pre-WES assessment allowed the authors to reframe the clinical suspect of GS in 13/36 (36%) patients referred for BS. Conversely, the clinical suspect of GS was confirmed by clinical evaluation in 19/19 (100%) patients. In patients referred for BS/GS, they refined the clinical suspect in BS in one patient and GS in 7 patients. Because the Authors performed WES analysis, they have the possibility to screen for all causative genes, so the question is: Why they re-evaluated the clinical suspicion before performing genetic testing? In other words, they coud have approached by “reverse phenotyping” (it seems that this is the approach then)

We recently developed a strategy for diagnosis of rare kidney diseases based on the close collaboration between peripheral nephrology and paediatric units and our tertiary referral centre. Patients are selected in peripheral centres based on simple, easy-to-apply clinical criteria (e.g., metabolic alkalosis and/or hypokalemia) and referred to the tertiary centre for genetic testing. Here, clinical data are re-evaluated in order to: 1. Confirm or not indication to genetic testing (i.e., excluding secondary causes or alternative diagnosis); 2. Assess clinical and laboratory data to optimise genetic results interpretation (i.e., providing the laboratory with accurate phenotypic information to guide variant prioritization, particularly regarding the pattern of inheritance and the clinical significance); 3. Provide patients and family with appropriate counselling about genetic testing procedure, addressing indications, expectations, limitations and psychological issues. In our manuscript, we aimed at describing this process that essentially reflects our daily clinical practice in order to highlight that the clinical suspicion of BS and GS can be misleading, especially for non-expert clinicians. Clinical re-evaluation before genetic testing allowed us to provide information about the rate of clinical diagnosis confirmed and modified by genetic testing. Indeed, although reverse genetics probably represents the future of genetic investigation strategies, detailed clinical phenotyping is pivotal in guiding variant prioritization, helping clinicians and biologists to assess the clinical significance of variant unknown significance.

The process of searching for pathogenic mutations generally begins with the sequencing of the known associated genes by NGS technologies. The GS and BS genes are often part of predesigned panels of known genes involved in tubulopathies, chronic hypokalemia or salt-acid disorders.
What was the approach of the authors? And did they evaluate the mitochondrial genome?

During the analysis of WES results, we proceed by performing in silico filtering for rare variants, according to the pattern of inheritance. We then perform prioritization analysis according to ACMG guidelines, as reported in Materials section. The analysis of medically actionable secondary findings is not performed, because the trial protocol and consent did not permit analysis of these patients’ data for secondary findings.

Unfortunately, we haven’t developed a strategy to evaluate mtDNA yet. Considering recent data (Viering D et al. JASN 2022) this surely represents an implementation to develop to improve our approach.

Another complementary method is the search for CNVs (copy number variations) in candidate genes, through specific NGS products and software, arrays of SNPs (single nucleotide polymorphism) or MLPA®(Multiplex Ligation-dependent Probe Amplification), among others. The authors investigated CNV through  NGS AND in house bioinformatic pipeline.
Did they confirm these CNV? B SNP Array or MLPA?

We confirmed all CNVs detected by WES by Real-time PCR or CGH-array, depending on the size of the deletion. In particular, we used CGH-array to confirm CLCNKB and/or CLCNKA homozygous and heterozygous deletions.

Round 2

Reviewer 1 Report

The authors have made major changes to the manuscript, but have not yet entered the ethics committee approval number. The manuscript cannot be accepted without the ethics committee.

Author Response

We added the approval number of local Ethic Committee in the Methods Section and in the Institutional Review Board Statement, as requested. We also revised the manuscript for typos and additional spell checks.
